# Prediction of child and adolescent outcomes with broadband and narrowband dimensions of internalizing and externalizing behavior using the child and adolescent version of the Strengths and Difficulties Questionnaire

**Pawel R. Kulawiak**[1,2]*, **Jürgen Wilbert**[1], **Robert Schlack**[3], **Moritz Börnert-Ringleb**[4]

**1** Department of Inclusive Education, University of Potsdam, Potsdam, Germany, **2** Department of Special Education and Rehabilitation, University of Cologne, Cologne, Germany, **3** Department of Epidemiology and Health Monitoring, Robert Koch Institute, Berlin, Germany, **4** Institute for Special Education, Leibniz University Hannover, Hannover, Germany

* kulawiak@uni-potsdam.de

## Abstract

The Strengths and Difficulties Questionnaire (SDQ) is a frequently used screening instrument for behavioral problems in children and adolescents. There is an ongoing controversy —not only in educational research—regarding the factor structure of the SDQ. Research results speak for a 3-factor as well as a 5-factor structure. The narrowband scales (5-factor structure) can be combined into broadband scales (3-factor structure). The question remains: Which factors (narrowband vs. broadband) are better predictors? With the prediction of child and adolescent outcomes (academic grades, well-being, and self-belief), we evaluated whether the broadband scales of internalizing and externalizing behavior (3-factor structure) or narrowband scales of behavior (5-factor structure) are better suited for predictive purposes in a cross-sectional study setting. The sample includes students in grades 5 to 9 ($N = 4642$) from the representative German Health Interview and Examination Survey for Children and Adolescents (KiGGS study). The results of model comparisons (broadband scale vs. narrowband scales) did not support the superiority of the broadband scales with regard to the prediction of child and adolescent outcomes. There is no benefit from subsuming narrowband scales (5-factor structure) into broadband scales (3-factor structure). The application of narrowband scales, providing a more differentiated picture of students' academic and social situation, was more appropriate for predictive purposes. For the purpose of identifying students at risk of struggling in educational contexts, using the set of narrowband dimensions of behavior seems to be more suitable.

## Introduction

Internalizing and externalizing behavior problems are considered a substantial risk factor for students' social and academic well-being [1]. Both dimensions are consistently associated with

**Data Availability Statement:** The data set of the survey cannot be made publicly available because informed consent from study participants did not cover public deposition of data. However, the minimal data set underlying the findings presented in this manuscript is archived in the Research Data Centre at the Robert Koch Institute (RKI) and can be accessed by all interested researchers. On-site access to the data set is possible at the Secure Data Center of the RKI's Research Data Centre. Requests should be submitted to fdz@rki.de.

**Funding:** The authors received no specific funding for this work.

**Competing interests:** The authors have declared that no competing interests exist.

difficulties on different levels in schools. Children and adolescents with externalizing and/or internalizing behavior problems are exposed to higher risks of academic failure [2–6]. In addition to academic difficulties, subtypes of externalizing as well as internalizing behavior problems are associated with a range of short- and long-term developmental risks, such as higher rates of social exclusion/rejection [7], school suspension [6], lack of bonding to school [7], or criminal arrest [8]. Therefore, the early and adequate identification of students at-risk for the development of severe externalizing and internalizing behavior problems in educational practice is of high relevance [9]. The gained insights in students' behaviors might at the same time serve as a reference for designing appropriate behavioral interventions. Teachers and school psychologists consequently play an important role in supporting and identifying students who suffer from behavioral problems.

Among others, the Strengths and Difficulties Questionnaire (SDQ) [10] is a frequently used screening instrument for behavioral and emotional problems in children and adolescents [11] and is commonly applied in educational research and practice [4, 12]. The SDQ comprises 25 items that can be grouped into subscales for emotional symptoms, peer problems, conduct problems, hyperactivity, and prosocial behavior, containing five items each [10]. The narrowband subscales for emotional symptoms and peer problems can be combined into the broadband scale *internalizing* behavior and the narrowband subscales of hyperactivity and conduct problems can be subsumed into the broadband scale *externalizing* behavior [13]. Correlations of behavioral and emotional problems, measured by means of sum scores on the SDQ, with academic outcome variables have been addressed in recent research. Previous studies have highlighted negative associations between the SDQ scales and academic achievement [4, 14–16]. Metsäpelto et al. [4, 15] found that higher externalizing behavior (broadband scale) was related to lower reading skills. Similarly, Palmu et al. [16] highlighted that externalizing behavior (broadband scale) was associated with lower academic grades. Similarly, DeVries et al. [17] concluded that peer problems (narrowband scale) are negatively associated with academic grades. Higher internalizing and externalizing behavior (broadband scales) were linked to lower scholastic performance [18]. Hyperactivity and conduct problems (narrowband scales) were negatively correlated with reading and mathematical skills [14]. Not only are the associations between the SDQ scales and academic performance of interest from a pedagogical standpoint, but also their relation to social outcomes, such as social integration within the classroom, the well-being of students in their families, school absenteeism, and feelings of self-worth and self-efficacy. Higher emotional symptoms, peer problems, conduct problems, and hyperactivity (narrowband scales) are associated with a lower social preference within the classroom and with more difficulties within the family [19] and lower feelings of self-worth [20]. Students rejected by peers are more likely to show internalizing behavior problems (broadband scale) and controversial students are more likely to exhibit externalizing behavior (broadband scale) [21]. Higher emotional symptoms (narrowband scale) are associated with lower feelings of self-efficacy [22]. Lenzen et al. [23] highlighted the association of greater conduct problems and emotional symptoms (narrowband scales) with increased school absenteeism. School absence was also linked to internalizing behavior (broadband scale) while school discipline referrals were related to externalizing behavior (broadband scale) [24]. Moreover, students with special educational needs, especially those with learning disabilities, have greater peer problems (narrowband scale) [12].

Although the originally proposed 5-factor structure of the SDQ (5 narrowband subscales) is often applied in research [12, 17, 19, 20, 22, 23], the narrowband subscales are sometimes subsumed into broadband behavior scales (internalizing and externalizing behavior) [4, 14, 15, 18, 21, 24], which has resulted in an ongoing controversy, not only in educational research, regarding the factor structure of the SDQ [25–27]. This controversy implies a discussion about the usefulness of broadband and narrowband scales of behavior.

## Broadband and narrowband scales of behavior

Some studies on the psychometric properties of the SDQ [13, 25, 26, 28, 29] have confirmed the original 5-factor structure (emotional symptoms, peer problems, conduct problems, hyperactivity, and prosocial behavior) proposed by Goodman [10]. At the same time, several studies have highlighted potential model fit shortcomings of the 5-factor structure [30–33]. As a reaction to these concerns, several authors have proposed a 3-factor model as a possible alternative to the original 5-factor model [13, 33]. In the revised 3-factor model, the narrowband subscales of emotional symptoms and peer problems are combined into the broadband scale for *internalizing* behavior, and the narrowband subscales of hyperactivity and conduct problems are subsumed into the broadband scale for *externalizing* behavior (the narrowband subscale of prosocial behavior remains unchanged). Recent research has provided partial support for the appropriateness of the revised 3-factor model. In studies comparing the different factor structures (3 vs. 5) in terms of model fit, the superiority of either the 3-factor structure [34–36] or the 5-factor structure [13, 37–39] or also the adequateness of both factor structures [13, 25, 28, 33] have been documented. Goodman et al. ([7] p. 1189) conclude that "*there may be no single best set of subscales to use in the SDQ; rather, the optimal choice may depend in part upon one's study population and study aims.*"

Subsuming narrowband subscales into broadband scales takes place from a clinical perspective and represents the well-known hierarchical model of child and adolescent psychopathology [40, 41]. For example, the SDQ narrowband scales of hyperactivity and conduct problems describe distinctive psychopathological phenomena but are often co-occurring [42], which is why both scales form a broadband externalizing scale. The broadband perspective on child and adolescent behavior and emotions leads to a two-dimensional taxonomy of psychopathology distinguishing between internalizing and externalizing behavior. The question of whether child and adolescent psychopathology is best described by narrowband or broadband measures is the subject of ongoing debate [43–45]:

> "[. . .] the once plausible goal of identifying homogeneous populations for treatment and research resulted in narrow diagnostic categories that did not capture clinical reality, symptom heterogeneity within disorders, and significant sharing of symptoms across multiple disorders. The historical aspiration of achieving diagnostic homogeneity by progressive subtyping within disorder categories no longer is sensible [. . .]" ([32] p. 12)

However, it must also be considered that different narrowband dimensions of behavior (e.g., aggressive vs. non-aggressive rule-breaking behavior) are related to different etiological factors [46]. Vice versa, different narrowband dimensions of behavior might explain educational outcome variables to varying degrees, for example, conduct problems are associated with arithmetic skills ($r$ = -.20), while the association is stronger for hyperactivity ($r$ = -.38) [14]. Subsuming these narrowband dimensions of behavior into a broader category of behavior might therefore lead to a loss of information; that is, differentiated effects (between narrowband scales) in predicting educational outcomes are not described by a broadband scale. The usage of narrowband scales could provide a nuanced description of the association between dimensions of behavior and child and adolescent outcomes.

## Aims

Empirical evidence concerning the superiority of the narrowband scales of behavior in the prediction of child and adolescent outcomes is lacking. Comparisons between the narrowband scales (5-factor structure) and broadband scales (3-factor structure) with regard to the

prediction of outcomes is sparse. The question remains: Which factors (narrowband vs. broadband) are better predictors? In addition, the vast majority of previous research using the SDQ has focused on parent- or teacher-reported student behavior. There is a lack of studies that examine the associations between self-reports of students on the SDQ and relevant outcomes. However, children and adolescents can be considered experts of their own well-being [47] and consequently might depict a valid and important source of information on their own behavior.

In the present study, therefore, we examined how behavioral and emotional problems, measured by means of the different self-report SDQ scores (narrowband and broadband scales), are associated with child and adolescent outcomes, such as measures of academic success (grades), well-being (school, friends, and family), and self-belief (self-esteem and self-efficacy). We thereby assume that narrowband scales of behavior are more informative predictors of outcomes than broadband scales of behavior. This comparison (narrowband vs. broadband dimensions of behavior) seems of particular importance for emphasizing the need to differentiate behavioral problems when examining associations with child and adolescent outcomes.

## Method

### Study design and participants

The analyzed cross-sectional sample was obtained from the baseline of the German Health Interview and Examination Survey for Children and Adolescents (KiGGS study) [48]. The KiGGS study is a nationally representative health survey comprising children and adolescents. The survey's main objective is to obtain information on key physical and mental health indicators, risk factors, health service utilization, health behavior, and living conditions of children and adolescents in Germany. Study participants were not recruited in schools (non-nested data structure) but randomly selected from the official registers of local residents. The data were collected from 2003 to 2006. The KiGGS sample consists of 17641 children and adolescents aged 0 to 17 years. The children were given a physical examination and the parents as well as the children and adolescents themselves (from age 11 on) were interviewed via written questionnaires. The study was approved by the Charité/Universitätsmedizin Berlin ethics committee and the Federal Office for the Protection of Data.

The present analysis represents a secondary data analysis with a focus on behavior/emotions and child and adolescent outcomes such as measures of well-being (school, friends, and family), academic success (grades), and self-belief (self-esteem and self-efficacy), which are factors of interest to professionals in educational contexts. We will focus on the child/adolescent-reported data and refer to children and adolescents of compulsory education age. Compulsory education in Germany usually ends with the completion of grade 9 (usually 15-year-old adolescents). The survey's questionnaires were addressed to children and adolescents from the age of 11 years onwards (usually children in grade 5 and above). The present sample therefore includes children and adolescents with a minimum age of 11 years in grades 5 to 9 ($N = 4642$; 52% boys; age in years: $M = 13.46$, $SD = 1.47$, $Min = 11.00$, $Q_1 = 12.17$, $Md = 13.42$, $Q_3 = 14.67$, $Max = 17.92$). The distribution of the children and adolescents across grades 6 to 9 is nearly equal (approximately 21.5% in each grade, but 14% in grade 5). The proportion of children and adolescents in grade 5 (usually 10- and 11-year-olds) is small, as individuals under the age of 11 are not included in the present sample.

### Measures

**Child and adolescent behavior and emotions.** The German self-report version of the Strengths and Difficulties Questionnaire (SDQ) was used to assess child and adolescent behavior and emotions [10, 49]. This questionnaire (25 items) quantifies emotional symptoms, peer

**Table 1. At-risk children and adolescents (SDQ: "Abnormal").**

| | *percent (frequency)* | | | *score range* | | |
|---|---|---|---|---|---|---|
| **measure** | **at-risk** | **borderline** | **normal** | **at-risk** | **borderline** | **normal** |
| emotional symptoms | 6.2% (286) | 6.3% (293) | 87.5% (4063) | $\geq 6$ | 5 | $\leq 4$ |
| peer problems | 6.5% (303) | 8.9% (415) | 84.5% (3924) | $\geq 5$ | 4 | $\leq 3$ |
| conduct problems | 5.3% (245) | 7.5% (348) | 87.2% (4049) | $\geq 5$ | 4 | $\leq 3$ |
| hyperactivity | 9.2% (428) | 9.2% (426) | 81.6% (3788) | $\geq 7$ | 6 | $\leq 5$ |
| internalizing behavior | 7.9% (369) | 4.5% (211) | 87.5% (4062) | $\geq 9$ | 8 | $\leq 7$ |
| externalizing behavior | 10.0% (467) | 6.0% (281) | 83.9% (3894) | $\geq 10$ | 9 | $\leq 8$ |

problems, conduct problems, hyperactivity, and, as a dimension of strength, prosocial behavior (original narrowband scales). Emotional symptoms and peer problems can be combined into a broadband internalizing behavior subscale, while conduct problems and hyperactivity can be subsumed into a broadband externalizing behavior subscale [13]. The prosocial behavior subscale was not considered in the present study, because the focus was on a comparison of the two broadband subscales with the underlying narrowband subscales. SDQ items are rated on a three-point scale (0 for "*not true*," 1 for "*somewhat true*," and 2 for "*certainly true*"). High subscale scores indicate elevated behavioral problems. The children and adolescents with the highest subscale scores, which are the upper 10% of the normative sample, can be categorized as "*abnormal*" and are considered to be at risk for psychiatric disorders [10, 50]. We therefore refer to these individuals as *at-risk* children and adolescents. A conservative classification rule [51], which minimizes false positive cases by selecting a cutoff value below 10%, was used to identify at-risk children and adolescents (emotional symptoms $\geq 6$, peer problems $\geq 5$, conduct problems $\geq 5$, hyperactivity $\geq 7$, internalizing behavior $\geq 9$, and externalizing behavior $\geq 10$; calculations based on KiGGS baseline data, Table 1). With regard to the self-report version used in the KiGGS study (data at hand), the internal consistencies (Table 2) for the subscales of peer problems and conduct problems are insufficient ($\alpha$ and $\omega \leq .50$), but moderate for the subscales of emotional symptoms, hyperactivity, and internalizing and externalizing behavior ($\alpha$ and $\omega \geq .60$).

**Child and adolescent outcomes.** *Health-related quality of life.* The self-report version of the KINDL-R is a brief questionnaire to measure the health-related quality of life of children and adolescents [52]. The subscales school (e.g., "doing my schoolwork was easy"), friends (e.g., "I played with friends"), self-esteem (e.g., "*I was proud of myself*"), and family (e.g., "*I got on well with my parents*") describe the students' well-being related to daily school life, friendship, family life, and feelings of self-worth. Each subscale consists of 4 items. Items are rated on a five-point scale (1 for "*never*," 2 for "*seldom*," 3 for "*sometimes*," 4 for "*often*," and 5 for "*all the time*"). High scores indicate a positive quality of life in the specific domain. With regard to the KiGGS study (data at hand), the internal consistencies (Table 2) of the mentioned subscales are mediocre ($\alpha$ and $\omega$ range from .53 to .69). The subscales physical and emotional well-being are not used in the present study.

*School grades.* The school grades (math and German) received on the last report card (half-year term) were reported by the parents. Germany uses a 6-point grading scale. School grades vary from 1 (excellent) to 6 (insufficient), which were reversed so higher values indicate a better academic performance.

*General self-efficacy.* The general self-efficacy scale is a 10-item questionnaire that was designed to assess optimistic self-beliefs in coping with a variety of difficult demands in life [53] (e.g., "*I can always manage to solve difficult problems if I try hard enough*"). In the KiGGS study, the scale was only used in adolescents aged 14 years and older ($N = 1750$). Items are

**Table 2. Correlations (Pearson's correlation), means, standard deviations, and reliabilities of SDQ subscales and child and adolescent outcome variables.**

| Measure | 1 | 2 | 3 | 4 | 5 | 6 | 7 | 8 | 9 | 10 | 11 | 12 | *M* | *SD* | *α*[b] | *ω*[b] |
|---|---|---|---|---|---|---|---|---|---|---|---|---|---|---|---|---|
| 1. emotional symptoms (SDQ) | | | | | | | | | | | | | 2.30 | 1.86 | .60 | .61 |
| 2. peer problems (SDQ) | **.31** | | | | | | | | | | | | 1.98 | 1.56 | .48 | .50 |
| 3. conduct problems (SDQ) | **.26** | **.20** | | | | | | | | | | | 1.95 | 1.39 | .43 | .46 |
| 4. hyperactivity (SDQ) | **.25** | **.11** | **.40** | | | | | | | | | | 3.74 | 2.04 | .64 | .69 |
| 5. internalizing behavior (SDQ) | **.85** | **.77** | **.29** | **.23** | | | | | | | | | 4.28 | 2.77 | .64 | .69 |
| 6. externalizing behavior (SDQ) | **.30** | **.17** | **.76** | **.90** | **.30** | | | | | | | | 5.69 | 2.88 | .67 | .62 |
| 7. school (KINDL-R) | **-.37** | **-.19** | **-.29** | **-.33** | **-.36** | **-.37** | | | | | | | 14.77 | 2.72 | .53 | .62 |
| 8. friends (KINDL-R) | **-.31** | **-.49** | **-.17** | **-.08** | **-.48** | **-.14** | **.25** | | | | | | 16.57 | 2.37 | .53 | .54 |
| 9. self-esteem (KINDL-R) | **-.20** | **-.16** | **-.14** | **-.17** | **-.23** | **-.19** | **.24** | **.27** | | | | | 13.09 | 2.96 | .67 | .69 |
| 10. family (KINDL-R) | **-.26** | **-.14** | **-.34** | **-.23** | **-.25** | **-.33** | **.35** | **.24** | **.21** | | | | 17.23 | 2.37 | .68 | .68 |
| 11. math grade | **-.11** | -.04 | **-.14** | **-.20** | **-.10** | **-.20** | **.29** | .02 | .07 | **.14** | | | 4.09 | 0.94 | - | - |
| 12. German grade | -.03 | **-.09** | **-.15** | **-.19** | -.07 | **-.21** | **.26** | .01 | .05 | **.10** | **.47** | | 4.12 | 0.85 | - | - |
| 13. general self-efficacy[a] | **-.27** | **-.25** | **-.13** | **-.23** | **-.32** | **-.23** | **.28** | **.32** | **.40** | **.17** | **.10** | .08 | 29.47 | 4.39 | .83 | .84 |

*Note.* All parameters are reported with regard to the raw data. Significant correlations ($p < .05$) are in **bold**.

[a]The scale was only used in adolescents aged 14 years and older ($N = 1750$).

[b]Reliability coefficients are Cronbach's $α$ and McDonald's $ω$.

rated on a four-point scale (1 for "*not at all true*," 2 for "*hardly true*," 3 for "*moderately true*," and 4 for "*exactly true*"). High scores indicate stronger self-efficacy. With regard to the KiGGS study (data at hand), the internal consistency (Table 2) of the scale is good ($α$ and $ω > .80$).

## Statistical analysis strategy

Ordinary least square regression models will be formulated with regard to the prediction of child and adolescent outcomes (outcomes regressed on SDQ subscales). The term "prediction" and cognate terms are used here in a statistical sense and shall not be confused with the concept of predictive validity, which describes the ability of a measure to forecast outcomes in the future [54]. To judge the statistical predictive performance of the different subscales of the SDQ (broadband vs. narrowband), the regression model with the broadband subscale (model 1: outcome regressed on broadband subscale, e.g., internalizing behavior) will be compared to the regression model with both underlying narrowband subscales jointly as predictors in one regression model (e.g., model 2: outcome regressed on emotional symptoms and peer problems). This model comparison (narrowband vs. broadband) will be conducted with regard to each predicted outcome and separately for internalizing and externalizing behavior (internalizing behavior vs. emotional symptoms and peer problems; externalizing behavior vs. conduct problems and hyperactivity). To evaluate the predictive performance of the different models (predictive performance of the broadband and narrowband subscales), we report two goodness-of-fit indices for each regression model. The adjusted $R^2$ is the proportion of variance in the outcome that is predictable from the predictors (SDQ subscales). The Akaike Information Criterion (AIC) takes into account both model complexity (total number of estimated model parameters) and goodness of model fit (maximized likelihood) and balances these two [55]. The individual AIC values are not interpretable. However, the smaller the AIC value, the better the model fit. Consequently, models with less complexity (fewer predictors) along with a high goodness of fit are deemed to be good models. If the difference in AIC values between the models is less than 3 (model with broadband scale vs. model with underlying narrowband scales), then the model with the higher AIC value is almost as good as the model with the

smaller AIC value [55]. For the application of AIC model selection in the fields of psychology and psychometrics, see Vrieze [56].

The SDQ subscales are used as dummy variables. The reference is the at-risk category ("abnormal"). Therefore, the intercept (constant) of each regression model is interpretable as the expected average outcome for the at-risk children and adolescents. Since all the outcomes are standardized ($M = 0$, $SD = 1$), the intercept represents the average outcome for the at-risk group as a deviation from the overall sample mean in units of standard deviation. The regression parameters ($B$) for all the other SDQ subscale scores are interpretable as the average difference in the outcomes (in units of standard deviation) between the at-risk group and the children and adolescents with the particular SDQ subscale score. These types of analyses emphasize the clinical category "abnormal" (at-risk). In some additional regression analyses, we will use the SDQ subscales as continuous predictors. If the SDQ subscale is a continuous predictor, the intercept represents the average outcome for children without behavioral problems (SDQ subscale score equals zero) as a deviation from the overall sample mean in units of standard deviation, and the regression parameter ($B$) is the average change (slope) of the outcome (in units of standard deviation) when the SDQ score increases on average by one unit. All statistical analyses were conducted in R 3.6.0.

## Results

### Preliminary results

**Model fit and measurement invariance of the 3- and 5-factor structures of the SDQ.** Confirmatory factor analyses (weighted least square mean and variance adjusted estimation) reveal an appropriate model fit ($RMSEA < 0.08$, for details see [57]) for both the 3- and 5-factor structures of the SDQ, although the 5-factor structure shows a better model fit ($RMSEA = .05$, $CFI = .89$, $TLI = .88$, $\chi^2 = 3250.87$, $df = 265$, $p = .00$) than the 3-factor structure ($RMSEA = .06$, $CFI = .85$, $TLI = .83$, $\chi^2 = 4540.10$, $df = 272$, $p = .00$). However, $CFI$ ($< .90$) and $TLI$ ($< .95$) do not indicate good fit for both the 3- and 5-factor structures. Both the 3- and 5-factor structures meet the standards for metric invariance [58, 59] across gender and age groups (multi-group confirmatory factor analysis; comparison of metric and configural model: difference in the models' CFI $\leq .01$), which can be interpreted to indicate that the measured dimensions of behavior (narrowband and broadband scales) manifest in the same way across boys and girls as well as different age groups (quartile age groups in years: [11,12.2], (12.2,13.4], (13.4,14.7], and (14.7,17.9]). As the goal of the present paper is to compare the statistical predictive performance of the broadband (3-factor structure) and narrowband (5-factor structure) scales of behavior and as it is not the goal to highlight differences between boys and girls or different age groups, sex and age are not considered as predictors of the child and adolescent outcomes.

### Descriptive results

Based on the SDQ narrowband subscales, the proportion of at-risk children and adolescents ranges between 5.3% and 9.2% (conduct problems: 5.3; emotional symptoms: 6.2; peer problems: 6.5; hyperactivity: 9.2%), while the SDQ broadband subscales reveal a proportion of 7.9 (internalizing behavior) and 10% (externalizing behavior) of at-risk children and adolescents (Table 1). The correlations, means, and standard deviations of the SDQ subscales and the child and adolescent outcomes are displayed in Table 2. All outcomes are positively associated. Increased positive correlations are observed between the grades in math and German ($r = .47$), as well as between the KINDL-R subscales of family and school ($r = .35$). The general self-efficacy scale is likewise considerably correlated with the KINDL-R subscales of self-esteem ($r = .40$) and friends ($r = .32$). The different SDQ subscales were negatively correlated

with all outcomes, which means that increased behavioral problems measured by means of the different SDQ subscales are associated with lower values for the outcomes, indicating adverse outcomes. The KINDL-R friends subscale is highly correlated with the SDQ subscales of peer problems ($r$ = -.49) and internalizing behavior ($r$ = -.48). The grades (math and German) were only weakly correlated with the SDQ subscales of peer problems and internalizing behavior ($r$ ranges from -.04 to -.10). Also, the correlation between the German grades and emotional symptoms is close to zero ($r$ = -.03). Another small correlation is between the KINDL-R subscale for friends and hyperactivity ($r$ = -.08).

## Main results: Predictive performance of the SDQ subscales

**Internalizing behavior vs. emotional symptoms and peer problems.**   Each child and adolescent outcome is regressed on the different SDQ subscales, which are the broadband subscale for internalizing behavior (model 1) and both underlying narrowband subscales, i.e., emotional symptoms and peer problems jointly as predictors in one regression model (model 2). The regression coefficients ($B$) and model fit parameters ($R^2$ and AIC) are displayed in Table 3.

With regard to the results of models 1 and 2, it can be stated that the associations between the outcomes and the SDQ subscales are negative, which means that increased behavioral problems as indicated by high SDQ subscale scores are associated with lower values of the outcomes, indicating adverse outcomes. Taken as a whole, the at-risk children and adolescents have the lowest average outcome values (the intercept ranges from -1.61 to -0.11).

With reference to the model fit parameters, the narrowband subscales of emotional symptoms and peer problems (model 2: jointly as predictors in one regression model) outperform the broadband subscale of internalizing behavior (model 1) in the prediction of all outcomes (comparable $R^2$ and lower AIC values), except for the prediction of general self-efficacy (AIC value favors the predictive performance of the broadband subscale).

However, the predictive performance of the narrowband and broadband scales is poor with regard to the prediction of the grades (math and German), i.e., the proportion of explained variance is close to zero ($R^2 \leq .01$). Therefore, it is hard to judge the predictive superiority of one of the SDQ subscales (with regard to grade prediction), although the AIC values favor the predictive performance of the narrowband subscales of emotional symptoms and peer problems (model 2).

Besides this, model 2 offers a deeper insight into the magnitude of the effect sizes of the two narrowband subscales. For example, in the prediction of the KINDL-R school subscale, the regression coefficients for emotional symptoms are remarkably higher ($B$ ranges from 0.37 to 1.30) than the coefficients for peer problems ($B$ ranges from 0.04 to 0.32). As well, in the prediction of the KINDL-R family subscale and in the prediction of the math grade, the emotional symptoms subscale shows noticeably higher coefficients than the peer problems subscale (family: $B$ ranges from 0.22 to 0.86 vs. 0.06 to 0.26, math: $B$ ranges from 0.04 to 0.34 vs. 0.03 to 0.06). The situation is reversed in the case of predicting the KINDL-R friends subscale, i.e., higher coefficients are observable for peer problems ($B$ ranges from 0.62 to 1.66), while lower coefficients are detected for emotional symptoms ($B$ ranges from 0.23 to 0.69). Also in the prediction of the German grade, the peer problems subscale shows higher coefficients than the emotional symptoms subscale ($B$ ranges from 0.02 to 0.30 vs. -0.10 to 0.00). This detailed information about the differences in effect sizes between the two narrowband subscales (model 2) is not depicted when the broadband subscale for internalizing behavior is used as a predictor (model 1).

The results are almost the same when the SDQ subscales are considered as continuous predictors (Table 4). With regard to the model fit parameters, the narrowband subscales of

**Table 3. Child and adolescent outcome variables regressed on SDQ subscales (internalizing behavior, emotional symptoms, peer problems).**

| | school KINDL-R | | friends KINDL-R | | self-esteem KINDL-R | | family KINDL-R | | math grade | | German grade | | general self-efficacy[b] | |
|---|---|---|---|---|---|---|---|---|---|---|---|---|---|---|---|
| | *B* | *SE* | *B* | *SE* | *B* | *SE* | *B* | *SE* | *B* | *SE* | *B* | *SE* | *B* | *SE* |
| **Model 1 (internalizing behavior)** | | | | | | | | | | | | | | |
| intercept[a] | -0.72 | 0.05 | -1.15 | 0.05 | -0.48 | 0.05 | -0.56 | 0.05 | -0.20 | 0.05 | -0.11 | 0.05 | -0.69 | 0.08 |
| score 8 | 0.21 | 0.08 | 0.52 | 0.08 | 0.18 | 0.08 | 0.11 | 0.08 | 0.03 | 0.09 | -0.10 | 0.09 | 0.27 | 0.13 |
| score 7 | 0.35 | 0.07 | 0.80 | 0.07 | 0.23 | 0.08 | 0.34 | 0.07 | 0.10 | 0.08 | 0.12 | 0.08 | 0.46 | 0.11 |
| score 6 | 0.51 | 0.07 | 0.87 | 0.06 | 0.40 | 0.07 | 0.45 | 0.07 | 0.16 | 0.07 | 0.06 | 0.07 | 0.49 | 0.11 |
| score 5 | 0.55 | 0.06 | 1.07 | 0.06 | 0.45 | 0.07 | 0.49 | 0.07 | 0.21 | 0.07 | 0.12 | 0.07 | 0.69 | 0.10 |
| score 4 | 0.74 | 0.06 | 1.21 | 0.06 | 0.46 | 0.06 | 0.58 | 0.06 | 0.18 | 0.07 | 0.05 | 0.07 | 0.64 | 0.10 |
| score 3 | 0.81 | 0.06 | 1.35 | 0.06 | 0.54 | 0.06 | 0.67 | 0.06 | 0.24 | 0.07 | 0.14 | 0.07 | 0.81 | 0.10 |
| score 2 | 1.09 | 0.06 | 1.53 | 0.06 | 0.70 | 0.06 | 0.87 | 0.06 | 0.22 | 0.07 | 0.10 | 0.07 | 0.94 | 0.10 |
| score 1 | 1.14 | 0.06 | 1.65 | 0.06 | 0.76 | 0.07 | 0.76 | 0.07 | 0.32 | 0.07 | 0.26 | 0.07 | 1.13 | 0.11 |
| score 0 | 1.40 | 0.08 | 1.94 | 0.08 | 0.87 | 0.09 | 0.97 | 0.08 | 0.41 | 0.09 | 0.31 | 0.09 | 1.56 | 0.14 |
| $R^2$/AIC | .13/12373 | | .22/11872 | | .05/12718 | | .07/12691 | | .01/12611 | | .01/12608 | | .11/4777 | |
| **Model 2 (emotional symptoms and peer problems)** | | | | | | | | | | | | | | |
| intercept[a] | -0.97 | 0.07 | -1.61 | 0.07 | -0.67 | 0.07 | -0.78 | 0.07 | -0.23 | 0.08 | -0.13 | 0.08 | -0.82 | 0.11 |
| *emotional symptoms* | | | | | | | | | | | | | | |
| score 5 | 0.37 | 0.08 | 0.23 | 0.07 | 0.29 | 0.08 | 0.22 | 0.08 | 0.04 | 0.09 | 0.00 | 0.09 | 0.12 | 0.12 |
| score 4 | 0.53 | 0.07 | 0.34 | 0.06 | 0.31 | 0.07 | 0.37 | 0.07 | 0.04 | 0.08 | -0.10 | 0.08 | 0.19 | 0.11 |
| score 3 | 0.67 | 0.07 | 0.31 | 0.06 | 0.47 | 0.07 | 0.53 | 0.07 | 0.14 | 0.07 | 0.00 | 0.07 | 0.32 | 0.10 |
| score 2 | 0.82 | 0.06 | 0.46 | 0.06 | 0.45 | 0.07 | 0.66 | 0.07 | 0.14 | 0.07 | -0.12 | 0.07 | 0.45 | 0.10 |
| score 1 | 1.04 | 0.06 | 0.56 | 0.06 | 0.59 | 0.07 | 0.75 | 0.07 | 0.23 | 0.07 | -0.02 | 0.07 | 0.56 | 0.10 |
| score 0 | 1.30 | 0.07 | 0.69 | 0.06 | 0.65 | 0.07 | 0.86 | 0.07 | 0.34 | 0.07 | -0.02 | 0.07 | 0.76 | 0.10 |
| *peer problems* | | | | | | | | | | | | | | |
| score 4 | 0.04 | 0.07 | 0.62 | 0.07 | 0.02 | 0.07 | 0.06 | 0.07 | 0.03 | 0.08 | 0.02 | 0.08 | 0.23 | 0.12 |
| score 3 | 0.07 | 0.06 | 0.98 | 0.06 | 0.10 | 0.07 | 0.17 | 0.07 | 0.10 | 0.07 | 0.09 | 0.07 | 0.26 | 0.11 |
| score 2 | 0.15 | 0.06 | 1.22 | 0.06 | 0.22 | 0.06 | 0.18 | 0.06 | 0.09 | 0.07 | 0.17 | 0.07 | 0.37 | 0.10 |
| score 1 | 0.21 | 0.06 | 1.41 | 0.06 | 0.28 | 0.06 | 0.25 | 0.06 | 0.07 | 0.07 | 0.23 | 0.07 | 0.48 | 0.10 |
| score 0 | 0.32 | 0.07 | 1.66 | 0.06 | 0.40 | 0.07 | 0.26 | 0.07 | 0.06 | 0.07 | 0.30 | 0.07 | 0.78 | 0.11 |
| $R^2$/AIC | .14/12296 | | .25/11674 | | .05/12708 | | .07/12667 | | .01/12600 | | .01/12598 | | .11/4783 | |

*Note*. All outcome variables are standardized (*M* = 0, *SD* = 1). The SDQ subscales are dummy coded.

[a]Reference: at-risk ("abnormal"), internalizing behavior $\geq$ 9, emotional symptoms $\geq$ 6, peer problems $\geq$ 5.

[b]The scale was only used in adolescents aged 14 years and older (*N* = 1750).

emotional symptoms and peer problems (model 2: jointly as predictors in one regression model) outperform the broadband subscale for internalizing behavior (model 1) in the prediction of all outcomes (comparable or lower $R^2$ and lower AIC values), except for the prediction of self-esteem and general self-efficacy (AIC values favor the predictive performance of the broadband subscale). Differences in effect sizes (*B*) between the two narrowband subscales (model 2) are apparent.

**Externalizing behavior vs. conduct problems and hyperactivity.** Each child and adolescent outcome is regressed on the different SDQ subscales, which are the broadband subscale for externalizing behavior (model 1) and both underlying narrowband subscales, i.e., conduct problems and hyperactivity jointly as predictors in one regression model (model 2). The regression coefficients (*B*) and model fit parameters ($R^2$ and AIC) are displayed in Table 5.

With regard to the results of models 1 and 2, it can be stated that the associations between the outcomes and the SDQ subscales are negative, which means that increased behavioral

**Table 4. Child and adolescent outcome variables regressed on SDQ subscales (continuous predictors: Internalizing behavior, emotional symptoms, peer problems).**

| | school KINDL-R | | friends KINDL-R | | self-esteem KINDL-R | | family KINDL-R | | math grade | | German grade | | general self-efficacy[a] | |
|---|---|---|---|---|---|---|---|---|---|---|---|---|---|---|
| | **B** | **SE** | **B** | **SE** | **B** | **SE** | **B** | **SE** | **B** | **SE** | **B** | **SE** | **B** | **SE** |
| **Model 1 (internalizing behavior)** | | | | | | | | | | | | | | |
| intercept | 0.55 | 0.03 | 0.74 | 0.02 | 0.35 | 0.03 | 0.39 | 0.03 | 0.15 | 0.03 | 0.11 | 0.03 | 0.52 | 0.04 |
| *internalizing behavior* | -0.13 | 0.00 | -0.17 | 0.00 | -0.08 | 0.01 | -0.09 | 0.01 | -0.04 | 0.01 | -0.03 | 0.01 | -0.12 | 0.01 |
| *$R^2$/AIC* | .13/12372 | | .23/11804 | | .05/12697 | | .06/12687 | | .01/12594 | | .01/12605 | | .11/4775 | |
| **Model 2 (emotional symptoms and peer problems)** | | | | | | | | | | | | | | |
| intercept | 0.54 | 0.03 | 0.76 | 0.02 | 0.35 | 0.03 | 0.38 | 0.03 | 0.14 | 0.03 | 0.12 | 0.03 | 0.52 | 0.04 |
| *emotional symptoms* | -0.18 | 0.01 | -0.09 | 0.01 | -0.09 | 0.01 | -0.13 | 0.01 | -0.06 | 0.01 | 0.00 | 0.01 | -0.11 | 0.01 |
| *peer problems* | -0.06 | 0.01 | -0.28 | 0.01 | -0.07 | 0.01 | -0.04 | 0.01 | -0.01 | 0.01 | -0.06 | 0.01 | -0.12 | 0.02 |
| *$R^2$/AIC* | .14/12290 | | .27/11594 | | .05/12698 | | .07/12654 | | .01/12586 | | .01/12590 | | .10/4777 | |

*Note.* All outcome variables are standardized ($M = 0$, $SD = 1$).

[a]The scale was only used in adolescents aged 14 years and older ($N = 1750$).

problems as indicated by high SDQ subscale scores are associated with lower values for the outcomes, indicating adverse outcomes. Taken as a whole, the at-risk children and adolescents have the lowest average outcome values (the intercept ranges from -0.93 to -0.18).

With regard to the model fit parameters, the narrowband subscales for conduct problems and hyperactivity (model 2: jointly as predictors in one regression model) outperform the broadband subscale of externalizing behavior (model 1) in the prediction of all outcomes (comparable or higher $R^2$ and lower AIC values).

In addition, model 2 offers a deeper insight into the magnitude of the effect sizes of the two narrowband subscales. For example, in the prediction of the KINDL-R friends subscale, the regression coefficients for conduct problems are remarkably higher (*B* ranges from 0.12 to 0.57) than the coefficients for hyperactivity (*B* ranges from -0.05 to 0.20). Similarly, in the prediction of the KINDL-R family subscale, the conduct problems subscale shows considerably higher coefficients (*B* ranges from 0.20 to 1.14) than the hyperactivity subscale (*B* ranges from 0.10 to 0.52). The situation is reversed in the case of predicting the math grade, i.e., higher coefficients are observable for hyperactivity (*B* ranges from 0.17 to 0.72), while lower coefficients are detected for conduct problems (*B* ranges from 0.18 to 0.31). In addition, in the prediction of general self-efficacy, the hyperactivity subscale shows noticeably higher coefficients (*B* ranges from 0.02 to 0.97) than the emotional symptoms subscale (*B* ranges from -0.07 to 0.33). This detailed information about the differences in effect sizes between the narrowband subscales (model 2) is not depicted when the broadband subscale for externalizing behavior is used as a predictor (model 1).

If the SDQ subscales are considered as continuous predictors (Table 6), the broadband subscale for externalizing behavior (model 1) outperforms the narrowband subscales (model 2) in the prediction of the KINDL-R subscales of school and self-esteem as well as in the prediction of the German grade (comparable $R^2$ and lower AIC values). In these predictions (KINDL-R school and self-esteem subscales as well as the German grade), there are no differences in effect sizes (*B*) between the two narrowband subscales, but they are apparent in the predictions of the other outcomes (KINDL-R subscales for friends and family, as well as math grades and general self-efficacy).

## Discussion

For the first time, the SDQ broadband and narrowband scales were compared with regard to their criterion validity in predicting child and adolescent outcomes. The results of the study

**Table 5. Child and adolescent outcome variables regressed on SDQ subscales (externalizing behavior, conduct problems, hyperactivity).**

| | school KINDL-R | | friends KINDL-R | | self-esteem KINDL-R | | family KINDL-R | | math grade | | German grade | | general self-efficacy[b] | |
|---|---|---|---|---|---|---|---|---|---|---|---|---|---|---|
| | **B** | **SE** | **B** | **SE** | **B** | **SE** | **B** | **SE** | **B** | **SE** | **B** | **SE** | **B** | **SE** |
| **Model 1 (externalizing behavior)** | | | | | | | | | | | | | | |
| intercept[a] | -0.65 | 0.04 | -0.18 | 0.05 | -0.27 | 0.05 | -0.69 | 0.04 | -0.36 | 0.05 | -0.34 | 0.05 | -0.24 | 0.07 |
| score 9 | 0.13 | 0.07 | -0.04 | 0.08 | 0.10 | 0.08 | 0.37 | 0.07 | 0.11 | 0.08 | 0.11 | 0.08 | -0.08 | 0.12 |
| score 8 | 0.38 | 0.06 | 0.09 | 0.07 | 0.14 | 0.07 | 0.47 | 0.06 | 0.20 | 0.07 | 0.18 | 0.07 | 0.02 | 0.10 |
| score 7 | 0.46 | 0.06 | 0.07 | 0.06 | 0.12 | 0.06 | 0.55 | 0.06 | 0.30 | 0.06 | 0.23 | 0.06 | 0.06 | 0.10 |
| score 6 | 0.62 | 0.06 | 0.20 | 0.06 | 0.22 | 0.06 | 0.68 | 0.06 | 0.34 | 0.06 | 0.31 | 0.06 | 0.26 | 0.09 |
| score 5 | 0.68 | 0.06 | 0.20 | 0.06 | 0.28 | 0.06 | 0.76 | 0.06 | 0.36 | 0.06 | 0.38 | 0.06 | 0.22 | 0.09 |
| score 4 | 0.85 | 0.06 | 0.27 | 0.06 | 0.36 | 0.06 | 0.94 | 0.06 | 0.44 | 0.06 | 0.46 | 0.06 | 0.31 | 0.10 |
| score 3 | 0.98 | 0.06 | 0.25 | 0.06 | 0.49 | 0.06 | 0.98 | 0.06 | 0.63 | 0.07 | 0.57 | 0.07 | 0.44 | 0.10 |
| score 2 | 1.16 | 0.07 | 0.44 | 0.07 | 0.54 | 0.07 | 1.10 | 0.07 | 0.59 | 0.07 | 0.63 | 0.07 | 0.65 | 0.11 |
| score 1 | 1.27 | 0.08 | 0.34 | 0.08 | 0.62 | 0.08 | 1.09 | 0.08 | 0.77 | 0.08 | 0.65 | 0.08 | 0.73 | 0.14 |
| score 0 | 1.61 | 0.11 | 0.73 | 0.11 | 0.89 | 0.11 | 1.45 | 0.11 | 0.80 | 0.11 | 0.85 | 0.11 | 1.35 | 0.20 |
| $R^2$/AIC | .14/12324 | | .02/12922 | | .04/12778 | | .11/12485 | | .04/12457 | | .04/12457 | | .06/4871 | |
| **Model 2 (conduct problems and hyperactivity)** | | | | | | | | | | | | | | |
| intercept[a] | -0.85 | 0.07 | -0.31 | 0.07 | -0.32 | 0.07 | -0.93 | 0.07 | -0.55 | 0.07 | -0.56 | 0.07 | -0.35 | 0.11 |
| *conduct problems* | | | | | | | | | | | | | | |
| score 4 | 0.17 | 0.08 | 0.12 | 0.08 | 0.03 | 0.08 | 0.20 | 0.08 | 0.18 | 0.08 | 0.14 | 0.08 | 0.08 | 0.12 |
| score 3 | 0.35 | 0.07 | 0.14 | 0.07 | 0.07 | 0.07 | 0.52 | 0.07 | 0.19 | 0.07 | 0.22 | 0.07 | -0.02 | 0.11 |
| score 2 | 0.45 | 0.07 | 0.23 | 0.07 | 0.06 | 0.07 | 0.75 | 0.07 | 0.25 | 0.07 | 0.30 | 0.07 | -0.07 | 0.10 |
| score 1 | 0.58 | 0.07 | 0.44 | 0.07 | 0.15 | 0.07 | 0.94 | 0.07 | 0.31 | 0.07 | 0.30 | 0.07 | 0.03 | 0.10 |
| score 0 | 0.79 | 0.07 | 0.57 | 0.08 | 0.37 | 0.08 | 1.14 | 0.08 | 0.31 | 0.08 | 0.45 | 0.08 | 0.33 | 0.12 |
| *hyperactivity* | | | | | | | | | | | | | | |
| score 6 | 0.08 | 0.06 | 0.00 | 0.07 | 0.03 | 0.07 | 0.10 | 0.06 | 0.17 | 0.07 | 0.20 | 0.07 | 0.02 | 0.11 |
| score 5 | 0.21 | 0.06 | 0.00 | 0.06 | 0.12 | 0.06 | 0.11 | 0.06 | 0.20 | 0.06 | 0.16 | 0.06 | 0.22 | 0.10 |
| score 4 | 0.34 | 0.06 | 0.02 | 0.06 | 0.24 | 0.06 | 0.19 | 0.06 | 0.28 | 0.06 | 0.27 | 0.06 | 0.30 | 0.10 |
| score 3 | 0.42 | 0.06 | -0.01 | 0.06 | 0.17 | 0.06 | 0.24 | 0.06 | 0.31 | 0.06 | 0.29 | 0.06 | 0.38 | 0.10 |
| score 2 | 0.61 | 0.06 | 0.01 | 0.06 | 0.31 | 0.06 | 0.27 | 0.06 | 0.44 | 0.06 | 0.40 | 0.06 | 0.42 | 0.10 |
| score 1 | 0.72 | 0.07 | -0.05 | 0.07 | 0.40 | 0.07 | 0.29 | 0.07 | 0.56 | 0.07 | 0.53 | 0.07 | 0.64 | 0.11 |
| score 0 | 1.00 | 0.08 | 0.20 | 0.08 | 0.55 | 0.08 | 0.52 | 0.08 | 0.72 | 0.08 | 0.61 | 0.08 | 0.97 | 0.14 |
| $R^2$/AIC | .14/12322 | | .03/12876 | | .04/12765 | | .13/12378 | | .04/12456 | | .04/12445 | | .07/4860 | |

*Note*. All outcome variables are standardized ($M = 0$, $SD = 1$). The SDQ subscales are dummy coded.

[a]Reference: at-risk ("abnormal"), externalizing behavior $\geq 10$, conduct problems $\geq 5$, hyperactivity $\geq 7$.

[b]The scale was only used in adolescents aged 14 years and older ($N = 1750$).

indicated the relevance of the different SDQ subscales for the description of students' socio-emotional and academic situation. At the same time, the results could not support a superiority of the broadband subscales with regard to prediction of the outcomes. This interpretation can be described for the internalizing and externalizing behavior subscales (except for the prediction of general self-efficacy, where the internalizing behavior scale shows the best model fit). If the SDQ scales are considered as continuous predictors, the broadband scale for internalizing behavior outperforms the narrowband scales in the prediction of general self-efficacy and self-esteem. The same holds true for the prediction of the KINDL-R subscales of school and self-esteem, as well as for the prediction of the German grade, where the continuous externalizing subscale shows the best model fit. At any rate, in all cases where a continuous

**Table 6. Child and adolescent outcome variables regressed on SDQ subscales (continuous predictors: Externalizing behavior, conduct problems, hyperactivity).**

| | school KINDL-R | | friends KINDL-R | | self-esteem KINDL-R | | family KINDL-R | | math grade | | German grade | | general self-efficacy[a] | |
|---|---|---|---|---|---|---|---|---|---|---|---|---|---|---|
| | **B** | **SE** | **B** | **SE** | **B** | **SE** | **B** | **SE** | **B** | **SE** | **B** | **SE** | **B** | **SE** |
| **Model 1 (externalizing behavior)** | | | | | | | | | | | | | | |
| intercept | 0.73 | 0.03 | 0.27 | 0.03 | 0.36 | 0.03 | 0.65 | 0.03 | 0.40 | 0.03 | 0.41 | 0.03 | 0.46 | 0.05 |
| *externalizing behavior* | -0.13 | 0.00 | -0.05 | 0.01 | -0.06 | 0.01 | -0.11 | 0.00 | -0.07 | 0.01 | -0.07 | 0.01 | -0.08 | 0.01 |
| *$R^2$/AIC* | .14/12318 | | .02/12923 | | .03/12777 | | .11/12462 | | .04/12445 | | .04/12433 | | .05/4878 | |
| **Model 2 (conduct problems and hyperactivity)** | | | | | | | | | | | | | | |
| intercept | 0.73 | 0.03 | 0.25 | 0.03 | 0.37 | 0.03 | 0.63 | 0.03 | 0.41 | 0.03 | 0.41 | 0.03 | 0.47 | 0.05 |
| *conduct problems* | -0.14 | 0.01 | -0.12 | 0.01 | -0.06 | 0.01 | -0.22 | 0.01 | -0.05 | 0.01 | -0.07 | 0.01 | -0.03 | 0.02 |
| *hyperactivity* | -0.12 | 0.01 | -0.01 | 0.01 | -0.07 | 0.01 | -0.06 | 0.01 | -0.08 | 0.01 | -0.07 | 0.01 | -0.11 | 0.01 |
| *$R^2$/AIC* | .14/12319 | | .03/12880 | | .03/12779 | | .13/12358 | | .04/12444 | | .04/12435 | | .05/4872 | |

*Note.* All outcome variables are standardized ($M = 0$, $SD = 1$).

[a]The scale was only used in adolescents aged 14 years and older ($N = 1750$).

broadband scale outperforms the underlying narrowband scales, the difference in AIC values between the models is less than 3 [55], i.e., the models with narrowband scales are almost as good as the models with the broadband scales.

The use of sum scores for the narrowband subscales of emotional symptoms and peer problems is more informative with respect to the range of predicted outcomes. The models using the narrowband scales indicate that there are differences between emotional symptoms and peer problems with regard to their effect on different outcome variables. This information is not depicted through the use of the broadband subscale of internalizing behavior, which might therefore lead to a loss of information. At the same time, it must be stated that children and adolescents might exhibit symptomatic behaviors related to emotional symptoms but not have major peer problems or vice versa [60]. Similar observations can be made for conduct problems and hyperactivity [61]. It can be assumed that different categories of behavioral problems (e.g. conduct problems and attention deficit/hyperactivity) also go in hand with the development of different outcomes over time [62, 63]. Similarly, different conditions and predictors might lead to either the one or the other narrowband behavior. Moreover, different developmental trajectories become clear when focusing different narrowband behaviors problems (e.g. emotional problems and peer problems) [60, 64, 65], Aggregating scores of narrowband into broadband scales might therefore run the risk of blending distinctive behaviors associated with different developmental outcomes. This evidence seems to strengthen the assumptions of Tandon et al. ([66] p. 593) who argue that:

> "[. . .] a major shift, and advance in this area, has been the study of more discrete differentiated disorders instead of lumping of all internalizing symptoms into one broad category of the two-dimensional internalizing versus externalizing taxonomy of childhood psychopathology."

Therefore, the differentiation between different narrow facets of internalizing behavior seems to be of particular importance in the description of child and adolescent psychopathology and the prediction of relevant outcomes. With respect to the broadband subscale of externalizing behavior, these assumptions can also be partially supported. In line with similar previous findings, differences between the narrowband subscales of conduct problems and hyperactivity with regard to their effect on the outcomes can also be described for most predictions.

## Limitations

In this context, however, it must be noted that the chosen outcome variables do not fully describe the levels of educational development of children and adolescents. Further research is desirable that applies in-depth assessment of educational outcomes, such as domain-specific academic achievement, social integration, cognitive or self-regulation processes and uses a multi-informant approach (especially data reported by educators and teachers are of relevance). Compared to school grades, a domain-specific assessment (e.g., reading comprehension) would provide a more detailed picture of the academic performance. Besides this, the reporting of school grades from the last report card by the parents might be prone to recall bias. In addition, some of the chosen outcome variables (KINDL-R scales school and friends) show only low internal consistencies and might therefore not be reliable. Unsatisfactory internal consistency can also be described with regard to some of the SDQ scales used (conduct problems and peer problems). However, it is important to note that poor reliability does not necessarily affect the goodness of predictive analysis [67].

## Conclusion

Despite the aforementioned limitations of the study at hand, the results shed light on the predictive abilities of different subscales of the SDQ. In addition to previous studies [13, 25, 28], these insights can be used for a further discussion of the advantages and disadvantages of different factor structures of the SDQ. In the sense of predicting educational outcomes, no advantage of the broadband scales (resulting from the 3-factor structure) become clear. The application of narrowband scales (resulting from the 5-factor structure), providing a more differentiated picture of the socio-emotional and academic situation of students, seems to be more appropriate for the prediction of child and adolescent outcomes. This interpretation is of course limited, as it refers to a selection of criteria and needs to be replicated with further educational outcome variables. Furthermore, future research should examine the validity of these results when using parent- or teacher-reported student behavior. In addition, the finding that differentiation of behavioral problems might be a benefit for the description of educational outcomes should be replicated using other behavior assessment tools than the SDQ. Nonetheless, for the purpose of identifying students at risk of struggling in educational contexts, using the set of narrowband dimensions of behavior seems to be more appropriate in educational research and practice.

## Implications

The study at hand indicated the need of focusing narrowband behaviors in educational practice in order to gain the most differentiated insights into possible predictors of the emotional-social as well as academic development of children and adolescents. It becomes clear, that different narrowband behaviors are more or less associated with different outcome variables. Subsuming behaviors into broadband categories in educational practice might lead to the fact that students, who might have been identified as at-risk because of salient narrowband behavior, might not be identified as at-risk in broadband categories (classification accuracy). In extension to these results of our analysis, the question arises, whether in educational practice, focusing narrowband behaviors might also be the most appropriate approach with regard to educational planning. Consequently, Casale et al. [68] argue, that the early identification of at-risk students and the provision of individualized intervention might be a key advantage of applying universal screening procedures (e.g. the SDQ) in schools. Gaining insights in specific behaviors might offer the most detailed information on educational needs, which might be

addressed in subsequent behavioral interventions. Subsuming behaviors might however lead to a loss of relevant information.

## Acknowledgments

We thank Jannis Bosch, Linda Kuhr, Anja Schwalbe and the reviewers for comments that greatly improved the manuscript.

## Author Contributions

**Conceptualization:** Pawel R. Kulawiak, Moritz Börnert-Ringleb.

**Formal analysis:** Pawel R. Kulawiak, Moritz Börnert-Ringleb.

**Supervision:** Jürgen Wilbert.

**Writing – original draft:** Pawel R. Kulawiak, Moritz Börnert-Ringleb.

**Writing – review & editing:** Pawel R. Kulawiak, Jürgen Wilbert, Robert Schlack, Moritz Börnert-Ringleb.

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
