## [Decision Letter · Decision Letter 0]

16 Jul 2020

PONE-D-20-15994

Prediction of School-Relevant Outcomes with Broadband and Narrowband Dimensions of Internalizing and Externalizing Behavior Using the Child and Adolescent Version of the Strengths and Difficulties Questionnaire

PLOS ONE

Dear Dr. Kulawiak,

Thank you for submitting your manuscript to PLOS ONE. After careful consideration, we feel that it has merit but does not fully meet PLOS ONE’s publication criteria as it currently stands. Therefore, we invite you to submit a revised version of the manuscript that addresses the points raised during the review process.

After thoroughly considering the reviews and reading the paper myself, I offer a number of points to consider in a potential revision, as well.

   1. Please enter a Financial Disclosure statement: The author(s) received no specific funding for this work.

   2. The use of the term “school relevant outcomes” for measures of subjective well-being, self-esteem and self-efficacy seems problematic. These constructs in themselves are the focus of important areas of research, therefore, for a multidisciplinary journal such as PLOS ONE, it would be preferable to use a wider term, for example “child outcomes” or “developmental outcomes”. I would also suggest to change the ambiguous term “social and personal factors” to more specific terms.

   3. What are reliabilities and cut-offs for the externalizing and internalizing scales?

   4. Please describe the KINDL-R in more detail: How many items are there? How many scales? What are the reliabilities?

   5. Grades. What time period is covered by the last report card?

   6. Please present the reliability of the general self-efficacy scale.

   7. Please provide chi square statistics and CFI for the CFA. Reviewer 2 also addresses this issue.

   8. Please provide the significance levels for the correlations in Table 2:

We look forward to receiving your revised manuscript.

Kind regards,

Helena R. Slobodskaya, M.D., Ph.D., D.Sc.

Academic Editor

PLOS ONE

Journal Requirements:

2.Thank you for stating the following financial disclosure:  [NO].

Reviewers' comments:

Reviewer's Responses to Questions

**Comments to the Author**

1. Is the manuscript technically sound, and do the data support the conclusions?

Reviewer #1: No

Reviewer #2: Yes

2. Has the statistical analysis been performed appropriately and rigorously? 

Reviewer #1: No

Reviewer #2: Yes

3. Have the authors made all data underlying the findings in their manuscript fully available?

Reviewer #1: No

Reviewer #2: Yes

4. Is the manuscript presented in an intelligible fashion and written in standard English?

Reviewer #1: Yes

Reviewer #2: Yes

5. Review Comments to the Author

Reviewer #1: The authors present an interesting question: how do the different factor-structures of the SDQ predict school grades. It is a question of interest to many educational researchers. However, it is difficult to assess how this question is answered based upon the unclear analyses used. It is necessary to further describe the methods in detail. It may be advised to conduct additional analyses to account for the complex data structure and to provide analyses that correspond more clearly to the goals of the article. Below are my more detailed comments:

Introduction:

1) The authors argue that the literature linking SDQ scales to academic achievement is sparse - there is research linking subscales of it to achievement, as well as SDQ to other related scales. Additional work describing some of these works may be useful for the reader. (e.g., DeVries, Rathmann, Gebhardt, 2018). Failing the specific comparisons of SDQ subscales to achievement, perhaps a discussion about the underlying constructs relationships to achievement could be expanded upon further.

2) In general the literature review is a bit brief, but to the point. It may be worth adding a section about predictive validity of the SDQ to other variables - who made what predictions and did they use narrow or broad scales?

Methods/Results:

3) Alpha and omegas in the .6 range while technically considered acceptable are still quite low - it may be worth making a cautionary note somewhere about this.

4) Also, why not give the exact values, as you do in other places (lines 170, 171, 173).

5) Alpha of for the KiGGS study is also very low, do you have the available omega? (line 191).

6) It appears you dropped prosocial behavior from all results. It may be worth a sentence somewhere explaining this and that it was done because it does not relate to the goals of the study.

7) It is altogether unclear exactly what analyses were done. Judging from the text and output tables, the authors appear to have run separate regressions for each dependent and independent variables. A multiple linear regression technique would be more appropriate here. It may be advised to run multiple linear models where the predictors are combined into the model, instead of a separate model for each predictor. Regardless, more information is required as the precise models that were developed and tested.

8) Is there no attempt to asses possible clusters and group effects? For this, a multilevel multiple regression may be advised.

Discussion

8) Without a clear picture to the exact analyses run, it is impossible to evaluate the validity of any conclusions made. The general argument appears to be that the more differentiated model(s?) has(have?) a better fit and moreover there is a variation in the betas for the subscores. This line of reasoning is possibly convincing, but it requires some additional details. can you provide some theoretical connection to this reasoning? The discussion itself is altogether brief, and would benefit from this as well as a deeper connection to previous work (which may be needed also in the Intro).

9) Some discussion of the predictive power of variables with low reliability may be relevant - either as a limitation or a caution. Essentially, the low reliability of the scales may have a major impact on predictive validity, which is a major focus of the article.

other notes:

line 84: provide to provided

Reviewer #2: The study entitled “Prediction of School-Relevant Outcomes with Broadband and Narrowband Dimensions of Internalizing and Externalizing Behavior Using the Child and Adolescent Version of the Strengths and Difficulties Questionnaire” is of great interest in the field of Child and Adolescent Psychological Health. The questionnaire SDQ is one of the most frequently used in this field and the approach is very stimulating. It contains new scientific knowledge and provides comprehensive information for further development in this productive line of research. This paper is well-argued and clearly worthy of publication.

It has several strengths, amongst others:

The background is adequate and up to date. The sample used is adequate and cross-sectional self-report data was obtained. Regression models have been compared with regard to the prediction of school-relevant outcomes using narrowband vs. broadband scales of behavior. The data are presented in a clear and easy-to-understand fashion. The results are clear and related to main goals. The conclusions are supported by data.

As minor comments I would like to say:

Introduction

The literature review could include some more recent research on the use of the SDQ questionnaire, as well as provide a stronger rational for the study of internalizing and externalizing behavior problems in this specific population.

Method

The age group is not clearly defined. It would be necessary to include frequencies and percentages of the age and sex distribution.

Results

In the result section, the authors informed that models for both 3 or 5 factors show good fit indices. Please include more than one indicator of model fit besides the RMSEA, such as Chi-square (χ2); degrees of freedom (df); p-value, CFI or TLI. In addition to this information, factor invariance of gender and age would be interesting to include in the analyses.

In table 1, in addition to the percentages of "abnormal" scores, it would be interesting to include those that are within the normal range and those that are at the limit.

Please, provide a better rational for choosing this type of regression and the AIC measure rather than, for example, other types of regression such as hierarchical regressions. Why did you test so many different regression models? Please clarify the purpose.

Discussion

In the discussion section, please describe in more detail the contribution of your study and its implications for practice in the educational context.

6. PLOS authors have the option to publish the peer review history of their article (what does this mean?). If published, this will include your full peer review and any attached files.

Reviewer #1: **Yes: **J M DeVries

Reviewer #2: **Yes: **Inmaculada Montoya-Castilla

---

## [Author Response · Author response to Decision Letter 0]

14 Sep 2020

Points raised by the academic editor

1. Please enter a Financial Disclosure statement: The author(s) received no specific funding for this work.

We added the statement to the current cover letter: “The author(s) received no specific funding for this work.”

2. The use of the term “school relevant outcomes” for measures of subjective well-being, self-esteem and self-efficacy seems problematic. These constructs in themselves are the focus of important areas of research, therefore, for a multidisciplinary journal such as PLOS ONE, it would be preferable to use a wider term, for example “child outcomes” or “developmental outcomes”. I would also suggest to change the ambiguous term “social and personal factors” to more specific terms.

We changed the term “school relevant outcomes” to “child and adolescent outcomes”. We removed the term “social and personal factors”. Instead, we use the term “self-belief”, for example (l. 157): “measures of academic success (grades), well-being (school, friends, and family), and self-belief (self-esteem and self-efficacy).”

3. What are reliabilities and cut-offs for the externalizing and internalizing scales?

Cut-offs (l. 206): “A conservative classification rule (37), which minimizes false positive cases by selecting a cutoff value below 10%, was used to identify at-risk children and adolescents (emotional symptoms ≥ 6, peer problems ≥ 5, conduct problems ≥ 5, hyperactivity ≥ 7, internalizing behavior ≥ 9, and externalizing behavior ≥ 10).” Reliability coefficients (Cronbach’s α and McDonald’s ω calculated with the data at hand) are reported in Table 2.

4. Please describe the KINDL-R in more detail: How many items are there? How many scales? What are the reliabilities?

Items (l. 230): “Each subscale consists of 4 items.”. Additional information about the subscales (l. 234): “The subscales physical and emotional well-being are not used in the present study.” Reliability coefficients (Cronbach’s α and McDonald’s ω calculated with the data at hand) are reported in Table 2.

5. Grades. What time period is covered by the last report card?

l. 235: “The school grades (math and German) received on the last report card (half-year term) were reported by the parents.”

6. Please present the reliability of the general self-efficacy scale.

Reliability coefficients (Cronbach’s α and McDonald’s ω calculated with the data at hand) are reported in Table 2.

7. Please provide chi square statistics and CFI for the CFA. Reviewer 2 also addresses this issue. 

Additional model fit parameters have been included (l. 307).

8. Please provide the significance levels for the correlations in Table 2.

Table note (Table 2): “Significant correlations (p < .05) are in bold.”

Reviewer #1

1. The authors argue that the literature linking SDQ scales to academic achievement is sparse - there is research linking subscales of it to achievement, as well as SDQ to other related scales. Additional work describing some of these works may be useful for the reader. (e.g., DeVries, Rathmann, Gebhardt, 2018). Failing the specific comparisons of SDQ subscales to achievement, perhaps a discussion about the underlying constructs relationships to achievement could be expanded upon further. 

The point we want to address is not that the literature that links SDQ scales to academic achievement is sparse, but that the literature that compares the narrowband and broadband scales in the prediction of outcomes is sparse. We added more information for clarification. We hope this clarifies your query. We have updated the literature to describe the issue in more detail: (Aanondsen et al., 2020; DeVries et al., 2018; Essau et al., 2012; Jones et al., 2020; McAloney-Kocaman & McPherson, 2017; Niclasen & Dammeyer, 2016; Zarrella et al., 2018)

2. In general the literature review is a bit brief, but to the point. It may be worth adding a section about predictive validity of the SDQ to other variables - who made what predictions and did they use narrow or broad scales?

We now make it clear (introduction) which studies used narrowband scales and which broadband scales.

3. Alpha and omegas in the .6 range while technically considered acceptable are still quite low - it may be worth making a cautionary note somewhere about this.

We updated the limitations (l. 502): “In addition, some of the chosen outcome variables (KINDL-R scales school and friends) show only low internal consistencies and might therefore not be reliable. Unsatisfactory internal consistency can also be described with regard to some of the SDQ scales used (conduct problems and peer problems). However, it is important to note that poor reliability does not necessarily affect the goodness of predictive analysis (67).”. 67 = Smits, N., van der Ark, L. A., & Conijn, J. M. (2018). Measurement versus prediction in the construction of patient-reported outcome questionnaires: Can we have our cake and eat it? Quality of Life Research, 27(7), 1673–1682. https://doi.org/10.1007/s11136-017-1720-4

4. Also, why not give the exact values, as you do in other places (lines 170, 171, 173).

Reliability coefficients (Cronbach’s α and McDonald’s ω calculated with the data at hand) are reported in Table 2.

5. Alpha of for the KiGGS study is also very low, do you have the available omega? (line 191).

Reliability coefficients (Cronbach’s α and McDonald’s ω calculated with the data at hand) are reported in Table 2.

6. It appears you dropped prosocial behavior from all results. It may be worth a sentence somewhere explaining this and that it was done because it does not relate to the goals of the study.

l. 199: “The prosocial behavior subscale was not considered in the present study, because the focus was on a comparison of the two broadband subscales with the underlying narrowband subscales.”

7. It is altogether unclear exactly what analyses were done. Judging from the text and output tables, the authors appear to have run separate regressions for each dependent and independent variables. A multiple linear regression technique would be more appropriate here. It may be advised to run multiple linear models where the predictors are combined into the model, instead of a separate model for each predictor. Regardless, more information is required as the precise models that were developed and tested.

Each outcome is predicted by the SDQ scales: Outcome regressed on broadband scale (Model 1) and outcome regressed on both underlying narrowband scales (Model 2). Hence, model 2 is a multiple linear model (two predictors: two narrowband scales). Both models are compared (AIC comparison) to judge the statistical predictive performance of the different subscales of the SDQ (broadband vs. narrowband), whereby this type of model comparison is conducted separately for internalizing and externalizing behavior (internalizing behavior vs. emotional symptoms and peer problems; externalizing behavior vs. conduct problems and hyperactivity). We reorganized the section “Statistical Analysis Strategy” (starts at l. 248) and added more information for clarification. We hope this clarifies your query.

8. Is there no attempt to asses possible clusters and group effects? For this, a multilevel multiple regression may be advised.

l. 171: “Study participants were not recruited in schools (non-nested data structure) but randomly selected from the official registers of local residents.” A multilevel regression is not necessary because the data structure is non-nested (random sample). 

9. Without a clear picture to the exact analyses run, it is impossible to evaluate the validity of any conclusions made. The general argument appears to be that the more differentiated model(s?) has(have?) a better fit and moreover there is a variation in the betas for the subscores. This line of reasoning is possibly convincing, but it requires some additional details. can you provide some theoretical connection to this reasoning? The discussion itself is altogether brief, and would benefit from this as well as a deeper connection to previous work (which may be needed also in the Intro).

Your summary of our research describes exactly the main point. We are glad that our main point is understandable. We added more information to highlight the research question and aims. In addition, we tried to address the need of a stronger theoretical connection as well as a stronger connection to previous work. Therefore, we extended the discussion with regard to previous findings on the independence of narrowband behaviors (ll.47 and 470) as well as the implications for educational planning (l. 520)

10. Some discussion of the predictive power of variables with low reliability may be relevant - either as a limitation or a caution. Essentially, the low reliability of the scales may have a major impact on predictive validity, which is a major focus of the article.

We updated the limitations (l. 502): “In addition, some of the chosen outcome variables (KINDL-R scales school and friends) show only low internal consistencies and might therefore not be reliable. Unsatisfactory internal consistency can also be described with regard to some of the SDQ scales used (conduct problems and peer problems). However, it is important to note that poor reliability does not necessarily affect the goodness of predictive analysis (67).” 67 = Smits, N., van der Ark, L. A., & Conijn, J. M. (2018). Measurement versus prediction in the construction of patient-reported outcome questionnaires: Can we have our cake and eat it? Quality of Life Research, 27(7), 1673–1682. https://doi.org/10.1007/s11136-017-1720-4

11. line 84: provide to provided 

Done

Reviewer #2

1. The literature review could include some more recent research on the use of the SDQ questionnaire, as well as provide a stronger rational for the study of internalizing and externalizing behavior problems in this specific population.

We have updated the literature: (Aanondsen et al., 2020; DeVries et al., 2018; Essau et al., 2012; Jones et al., 2020; McAloney-Kocaman & McPherson, 2017; Niclasen & Dammeyer, 2016; Zarrella et al., 2018). We see the need of providing a stronger rational for the study of internalizing and externalizing behavior in this specific population. Therefore, we revised parts of the theoretical background (l. 47) and added some references emphasizing the need and rational for the study of internalizing and externalizing behavior problems in this specific population.

2. The age group is not clearly defined. It would be necessary to include frequencies and percentages of the age and sex distribution.

Minimum, maximum and quartiles have been included to describe the distribution (age), l. 185: “The present sample therefore includes children and adolescents with a minimum age of 11 years in grades 5 to 9 (N = 4654; 52% boys; age in years: M = 13.46, SD = 1.47, Min = 11.00, Q1 = 12.17, Md = 13.42, Q3 = 14.67, Max = 17.92).”

3. In the result section, the authors informed that models for both 3 or 5 factors show good fit indices. Please include more than one indicator of model fit besides the RMSEA, such as Chi-square (χ2); degrees of freedom (df); p-value, CFI or TLI. In addition to this information, factor invariance of gender and age would be interesting to include in the analyses. 

Additional model fit parameters have been included (l. 307). Factor invariance (also referred to as measurement invariance): “Both the 3- and 5-factor structures meet the standards for metric invariance (47,48) across gender and age groups (multi‑group confirmatory factor analysis; comparison of metric and configural model: difference in the models’ CFI ≤ .01), which can be interpreted to indicate that the measured dimensions of behavior (narrowband and broadband scales) manifest in the same way across boys and girls as well as different age groups (quartile age groups in years: [11,12.2], (12.2,13.4], (13.4,14.7], and (14.7,17.9]).” (l. 309)

4. In table 1, in addition to the percentages of "abnormal" scores, it would be interesting to include those that are within the normal range and those that are at the limit.

Done (see Table 1)

5. Please, provide a better rational for choosing this type of regression and the AIC measure rather than, for example, other types of regression such as hierarchical regressions. Why did you test so many different regression models? Please clarify the purpose.

We have added an important source: “For the application of AIC model selection in the field of psychology and psychometrics, see Vrieze (53).” (l. 271). Vrieze, S. I. (2012). Model selection and psychological theory: A discussion of the differences between the Akaike information criterion (AIC) and the Bayesian information criterion (BIC). Each outcome is predicted by the SDQ scales: Outcome regressed on broadband scale (Model 1) and outcome regressed on both underlying narrowband scales (Model 2). Hence, model 2 is a multiple linear model (two predictors: two narrowband scales). Both models are compared (AIC comparison) to judge the statistical predictive performance of the different subscales of the SDQ (broadband vs. narrowband), whereby this type of model comparison is conducted separately for internalizing and externalizing behavior (internalizing behavior vs. emotional symptoms and peer problems; externalizing behavior vs. conduct problems and hyperactivity). We reorganized the section “Statistical Analysis Strategy” (starts at l. 248) and added more information for clarification. We hope this clarifies your query.

6. In the discussion section, please describe in more detail the contribution of your study and its implications for practice in the educational context.

We have added a subsection on the “Implications” of our study (l. 520 discussing the advantages of applying broad- or narrowband categorizations of behavior in schools with regard to diagnostic practice and educational planning.

---

## [Decision Letter · Decision Letter 1]

24 Sep 2020

Prediction of Child and Adolescent Outcomes with Broadband and Narrowband Dimensions of Internalizing and Externalizing Behavior Using the Child and Adolescent Version of the Strengths and Difficulties Questionnaire

PONE-D-20-15994R1

Dear Dr. Kulawiak,

We’re pleased to inform you that your manuscript has been judged scientifically suitable for publication and will be formally accepted for publication once it meets all outstanding technical requirements.

Kind regards,

Helena R. Slobodskaya, M.D., Ph.D., D.Sc.

Academic Editor

PLOS ONE

Additional Editor Comments (optional):

Reviewers' comments:

Reviewer's Responses to Questions

**Comments to the Author**

1. If the authors have adequately addressed your comments raised in a previous round of review and you feel that this manuscript is now acceptable for publication, you may indicate that here to bypass the “Comments to the Author” section, enter your conflict of interest statement in the “Confidential to Editor” section, and submit your "Accept" recommendation.

Reviewer #1: All comments have been addressed

2. Is the manuscript technically sound, and do the data support the conclusions?

Reviewer #1: Yes

3. Has the statistical analysis been performed appropriately and rigorously? 

Reviewer #1: Yes

4. Have the authors made all data underlying the findings in their manuscript fully available?

Reviewer #1: Yes

5. Is the manuscript presented in an intelligible fashion and written in standard English?

Reviewer #1: Yes

6. Review Comments to the Author

Reviewer #1: All recommended changes have been addressed. The exact analyses and procedures are now quite clear, as well as the conclusions and limitations.

7. PLOS authors have the option to publish the peer review history of their article (what does this mean?). If published, this will include your full peer review and any attached files.

Reviewer #1: No

---

## [Editor Report · Acceptance letter]

29 Sep 2020

PONE-D-20-15994R1 

Prediction of child and adolescent outcomes with broadband and narrowband dimensions of internalizing and externalizing behavior using the child and adolescent version of the strengths and difficulties questionnaire 

Dear Dr. Kulawiak:

I'm pleased to inform you that your manuscript has been deemed suitable for publication in PLOS ONE. Congratulations! Your manuscript is now with our production department. 

Kind regards, 

on behalf of

Dr. Helena R. Slobodskaya 

Academic Editor

PLOS ONE